# Conduction Mechanisms in Au/0.8 nm–GaN/n–GaAs Schottky Contacts in a Wide Temperature Range

**DOI:** 10.3390/ma14205909

**Published:** 2021-10-09

**Authors:** Hicham Helal, Zineb Benamara, Mouhamed Amine Wederni, Sabrine Mourad, Kamel Khirouni, Guillaume Monier, Christine Robert-Goumet, Abdelaziz Rabehi, Arslane Hatem Kacha, Hicham Bakkali, Lionel C. Gontard, Manuel Dominguez

**Affiliations:** 1Laboratoire de Microélectronique Appliquée, Université de Sidi Bel Abbès BP 89, Sidi Bel Abbes 22000, Algeria; benamara20022000@yahoo.fr (Z.B.); rab_ehi@hotmail.fr (A.R.); arslane_k@hotmail.com (A.H.K.); 2Department of Condensed Matter Physics and IMEYMAT, Institute of Research on Electron Microscopy and Materials, University of Cádiz, Campus Universitario de Puerto Real, E11510 Cádiz, Spain; lionel.cervera@gm.uca.es (L.C.G.); manolo.dominguez@uca.es (M.D.); 3Unité de recherche Matériaux Avancés et Nanotechnologies (URMAN), Institut Supérieur des Sciences Appliquées et de Technologie de Kasserine, Université de Kairouan, BP 471, Kasserine 1200, Tunisia; wederni.mohamed89@gmail.com; 4Laboratory of Physics of Materials and Nanomaterials Applied to the Environment, Faculty of Sciences of Gabes, University of Gabes, Gabès 6079, Tunisia; sabrine.skmc@gmail.com (S.M.); kamel.khirouni@fsg.rnu.tn (K.K.); 5Institut Pascal, CNRS, SIGMA Clermont, Université Clermont Auvergne, F-63000 Clermont-Ferrand, France; guillaume.monier@uca.fr (G.M.); christine.robert-goumet@uca.fr (C.R.-G.); 6Department of Material Science, Metallurgical Engineering and Inorganic Chemistry and IMEYMAT, Institute of Research on Electron Microscopy and Materials, University of Cádiz, Campus Universitario de Puerto Real, E11510 Cádiz, Spain; hicham.bakkali@gm.uca.es

**Keywords:** nitridation, GaN/n–GaAs, Schottky diode, I-V-T, conduction mechanisms, barrier height

## Abstract

Au/0.8 nm–GaN/n–GaAs Schottky diodes were manufactured and electrically characterized over a wide temperature range. As a result, the reverse current *I_inv_* increments from 1 × 10^−7^ A at 80 K to about 1 × 10^−5^ A at 420 K. The ideality factor *n* shows low values, decreasing from 2 at 80 K to 1.01 at 420 K. The barrier height *q*ϕ_*b*_ grows abnormally from 0.46 eV at 80 K to 0.83 eV at 420 K. The tunnel mechanism TFE effect is the responsible for the *q*ϕ_*b*_ behavior. The series resistance *R_s_* is very low, decreasing from 13.80 Ω at 80 K to 4.26 Ω at 420 K. These good results are due to the good quality of the interface treated by the nitridation process. However, the disadvantage of the nitridation treatment is the fact that the GaN thin layer causes an inhomogeneous barrier height.

## 1. Introduction

Metal-semiconductor (MS) contacts are very important in microelectronics [1,2,3,4]. They are used in optoelectronic devices, bipolar integrated circuits, high-temperature, and high-frequency applications [5,6]. The thermionic emission (TE) theory is the principal theory used to determine the parameters of the Schottky contact. 

However, the experimental current-voltage (I-V) characteristics present some anomalies at low temperatures, and both the Schottky barrier height and ideality factor are temperature-dependent [5,7,8,9,10]. 

This deviation of the thermionic emission theory is corrected by introducing other mechanisms, operating at the Schottky barrier such as the thermionic field emission TFE and the emission field FE currents [5,11].

The origin of these currents is explained by considering several phenomena. First, the Schottky barrier is typically not homogeneous in space [7,12,13,14,15], as measured using ballistic electron emission microscopy [16,17]. The most widely accepted approach for interpreting experimental data considers that the spatial barrier inhomogeneity can be modeled with a Gaussian distribution function [12,18,19,20,21,22,23,24]. 

Secondly, the existence of interface states [7,25] act as recombination centers and generate local electric fields, causing random metallic paths, reducing carrier lifetime, and inducing a large leakage current [26,27,28,29]. These interface states come from surface dislocations and surface contaminations incorporated during the elaboration process [27,30,31]. In addition, the Schottky metallization step can cause interfacial modifications [31,32,33,34]. 

Therefore, the interface quality has an essential impact on device behavior and performance. In this context, surface passivation is the best method of controlling the defective states [27,30,31,35,36,37,38,39]., Many studies on the nitridation of the GaAs surface have been carried out [27,30,37,38,40,41,42,43] to improve the behavior and the electrical properties of the Schottky contacts (e.g., the ideality factor, barrier height, saturation current, series resistance and reverse current. Moreover, the nitride layers have good stability against the formation of amorphous surface oxides, high electronegativity, and thermal stability [27,44]. 

In this work, we measure the electrical characteristics of Au/0.8 nm–GaN/n–GaAs Schottky contacts fabricated by using a glow-discharge plasma source (GDS) for nitridation. Moreover, we analyze the current transport mechanisms, and several electrical parameters are characterized in a wide range of temperatures (80–420 K).

## 2. The Experiment

The Schottky contacts were elaborated using commercially available Si-doped n-GaAs (100) substrates, of a thickness of 400 µm and an electron concentration *N_d_* = 4.9 × 10^15^ cm^−3^. The samples were cleaned chemically using H_2_SO_4_, deionized water, cold and hot methanol sequentially and dried with N_2_. Then, the surfaces were bombarded with Ar+ ions of about 1 keV (a sample current equal to 5 μA cm^−2^ during 1 h) in UHV conditions [30,40]. After surface cleaning, the substrates were heated at 500 °C and nitrided using a glow discharge nitrogen plasma source, running at 5 W for 30 min in a UHV chamber (Institut Pascal, Clermont-Ferrand, France). This nitridation process led to the growth of a 0.8 nm-thick layer of undoped GaN. Following the nitridation step, the samples were annealed at 620 °C for 1 h to crystallize the GaN layer [39,45,46]. 

A XPS system characterized by a dual anode Al–Mg X-ray source (Institut Pascal, Clermont-Ferrand, France) and hemispherical electron energy analyzer (Institut Pascal, Clermont-Ferrand, France) were used for the in situ measurement of the chemical composition and crystal structure. The GaN thickness was calculated by comparing the experimental spectra data to the theoretical XPS peak intensities and positions [38]. The Au dots were deposited with area of 4.41 × 10^−3^ cm^2^ and thickened to 100 nm. A Bruker Dimension Icon atomic force microscope (AFM, Bruker, Cádiz, Spain) equipped with ScanAsyst and Nanoscope software 9.7 (ScanAsyst, Cádiz, Spain) was used to investigate the film surface roughness. Using the PeakForce tapping mode, AFM topography measurements were taken in the air. To accomplish this, a silicon tip on a nitride cantilever (ScanAsyst Air model, Cádiz, Spain), with a 0.4 N m^−1^ spring constant and a nominal tip radius of 2 nm were used to examine regions of 1 × 1 μm^2^ with a resolution of 256 × 256 pixels. The current–voltage measurements were investigated under different temperatures (80–420 K), by a current source Keithely 220 (Laboratory of Physics of Materials and Nanomaterials Applied to the Environment, Gabès, Tunisia).

## 3. The Results

Figure 1 presents the PeakForce tapping AFM topography images for (a) the GaN surface and (b) an Au electrode within a 1 × 1 μm^2^ scan area represented at the same height scale. The Au texture was formed by interconnected grain channels, while that of GaN was almost flat. The root mean square (RMS) surface roughness values indicated a difference of almost one order of magnitude between Au and GaN smoothness, i.e., 0.3 nm for GaN and 4.6 nm for Au. The value of the RMS surface roughness for GaN was calculated by neglecting the areas occupied by the contamination features that clearly stand out in the Figure 1a. This value of roughness (0.3 nm) is less than half the nominal GaN layer thickness (0.8 nm).

The roughness difference is better shown in Figure 1c where the frontier between the Au electrode and the GaN surface is shown as a rendered illuminated 3D AFM image, and the different topographies of Au and GaN are clearly shown. To bring out this difference in roughness more clearly, the height distribution histograms shown in Figure 1d were obtained from the topography images. GaN exhibits a narrow peak, showing that in comparison to Au, which has a larger peak, the surface layer is more homogeneous. To clarify the difference between the peaks, they have been fitted to a single Gaussian distribution with a peak centered at 2.15, and 21.3 nm, and a full width at half maximum (FWHM) of 1.3 and 9.5 nm for GaN and Au, respectively.

Surface roughness induces a non-uniformity of thickness, a distribution of interfacial charges, and a local variation of the Fermi level. These phenomena yield to the inhomogeneity of the Schottky barrier height and affect the transport mechanism [47].

Figure 2 depicts the I-V characteristics of the Au/0.8 nm–GaN/n–GaAs structure, at temperatures ranging from 80 to 420 K. 

The values of the reverse current *I_Rev_* at −1 V and the threshold voltage *V_Th_* were extracted and illustrated in Figure 3. With increasing temperature, *I_Rev_* increased exponentially from 1 × 10^−7^ A at 80 K to 1 × 10^−5^ A at 420 K, and *V_Th_* decreased from 0.65 V at 80 K to 0.2 V at 420 K. 

The expression of the current for non-ideal Schottky diodes is [48]:(1)$$I=I_{s}\left(exp\left(\frac{q\left(V-IR_{s}\right)}{nkT}\right)-1\right)$$
and
(2)$$I_{s}=AA^{*}T^{2}\;\left(-\frac{q\phi_{b}}{kT}\right)$$where, $I_{s}$ is the saturation current; $R_{s}$ is the series resistance; $q\phi_{b}$ is the barrier height; *n* is the ideality factor; *k* is the Boltzmann constant; *A* is the effective diode area, and$\;A^{*}$ is the effective Richardson constant equal to 8.16 A${\mathrm{cm}}^{-2}K^{2}$ for GaAs.

At the low bias voltage *V*, the current *I* is low, therefore the term $IR_{s}$ is low compared to *V*, and (Equation (1)) becomes
(3)$$I=I_{s}exp\left(\frac{qV}{nkT}\right)$$and
(4)$$ln(I)\;=\;\frac{q}{nkT}V+ln(I_{s})$$

The *n* and $I_{s}$ values are calculated from the slope and *y*-intercept of *ln(I)*-*V*, respectively. The$\;\phi_{b}$ value is determined as follows:(5)$$\phi_{b}=\frac{kT}{q}ln\left(\frac{AA^{*}T^{2}}{I_{s}}\right)$$

The *R_s_* values are extracted using the Cheung and Cheung method [48] which is based on
(6)$$G(I)\;=\;\frac{\partial V}{\partial \left(lnI\right)}\;=\;R_{s}I+\frac{nkT}{q}$$

The extracted values of *n* and $q\phi_{b}$ are plotted in Figure 4. 

As can be seen from Figure 4, with the rising temperature, n dropped from 2 for 80 K to 1.1 for 420 K. The decrease was very slow from 250 K to 450 K, which is in accordance with the literature [5,6,7,11,26]. The low values of n may have been due to the effect of the nitridation process, which improves the quality of the interface. As the temperature rose, $q\phi_{b}$ rose abnormally from 0.46 eV for 80 K to 0.83 eV for 420 K. These results were similar to several studies [7,12,20,49,50,51]. For Schottky contacts, the $q\phi_{b}$ value should decrease as the temperature rises, due to the bandgap’s temperature variation [1,2,7,48,50,52,53,54]. The $q\phi_{b}$ behavior may be explained by tunnel effect mechanisms, such as thermionic field emission (TFE) [5,11]. 

The tunneling current can be expressed following [1,12,55,56] as
(7)$$I=\;I_{tun}\left[exp\left(\frac{q(V-IR_{s}}{E_{0}}\right)-1\right]$$
(8)$$\frac{E_{0}}{kT}\;=\frac{E_{00}}{kT}cot\;h\left(\frac{E_{00}}{kT}\right)\;$$
(9)$$E_{00}=\frac{h}{4\pi }{\left(\frac{N_{D}}{m_{e}^{*}\varepsilon_{s}}\right)}^{\frac{1}{2}}$$where *E*_00_ is the characteristic tunneling energy; *h* is the Planck constant; $m_{e}^{*}$ is the effective mass of electron; and $\varepsilon_{s}$ is the dielectric constant of GaAs. Figure 5 shows the variation of (*E*_0_
*= nkT/q)* versus *kT/q*.

From Figure 5, *E*_0_ is about *kT/q*, which confirms that the TFE mechanism is dominant [26], not the theoretical mechanism TE of the Schottky contacts. This explains the abnormal behavior of the barrier height and the deviation of the ideality factor from unity. This may have been due to the interface states, which behaved as recombination–generation centers that affected the conduction mechanism [57].

To further study the abnormal behavior of the barrier height, the Richardson characteristic *ln*($I_{s}/T^{2}$) versus *q/kT* is presented in Figure 6 using the equation
(10)$$ln\left(\frac{I_{s}}{T^{2}}\right)\;=\;ln(AA^{*})-q\frac{\;\phi_{b_{n}}}{kT}$$

Figure 6 gives two linear regions which are due to the inhomogeneity of the barrier height [12]. $q\phi_{b}$ and $A^{*}$ values are 1.02 eV and 4.15 × 10^3^ A${\mathrm{cm}}^{-2}K^{-2}$ respectively in region 1 and equal to 0.19 eV and 3.6 × 10^21^ A${\mathrm{cm}}^{-2}K^{-2}$ respectively in region 2. These values of $A^{*}$ are significantly far from the theoretical value 8.16 A${\mathrm{cm}}^{-2}K^{-2}$ for n-GaAs [52].

Figure 7 presents the variation of $\phi_{b}$ versus *n*.

The structure has two linear characteristics due to barrier height inhomogeneity [58,59]. By extrapolation, the estimated values of $q\phi_{b}$ for *n* = 1 are 0.87 eV for region 1, and 0.84 eV for region 2. These values are closer than those extracted from the Richardson characteristics. 

The authors of this work [5] previously performed simulations of Au/n–GaAs Schottky at temperatures ranging from 80 to 400 K, with and without a thin GaN (1 nm) interfacial layer. They found that Au/n–GaAs shows a homogeneous barrier height while Au/1 nm–GaN/n–GaAs structure shows an inhomogeneous one. Therefore, the experimental results shown here––the inhomogeneity of the barrier height shown in the Richardson characteristics and in the plot of $q\phi_{b}$ versus *n*––are most likely because of the 0.8 nm GaN layer. Figure 8 illustrates *G(I)* plots of the Cheung and Cheung method at temperatures 80–420 K. 

*R_s_* and *n* were extracted by the Cheung and Cheung method for each temperature and presented in Figure 9 and Figure 10, respectively.

As can be seen, the structure gives the low resistance series *R_s_*, which decreased from 13.80 Ω at 80 K to 4.26 Ω at 420 K, showing the good quality of the interface improved by nitridation and annealing [28]. 

The *n* values were very high at low temperatures compared to those extracted from the first method. This discrepancy occurred because the *n* values obtained by the first method were extracted from the low bias voltage range, where the series resistance is very low. On the other hand, the *n* values extracted using the Cheung and Cheung method were extracted from all bias voltage ranges, where the series resistance in high bias voltages affects the calculation of the ideality factor.

Finally, the growth of a 0.8 nm of GaN layer on n-GaAs surfaces with an annealing process led to improved electrical parameters of the Schottky contacts, such as the series resistance and the ideality factor. However, it can cause the inhomogeneity of the barrier height at the structure.

## 4. Conclusions

Au/0.8 nm–GaN/n–GaAs structures were fabricated using a glow discharge plasma source (GDS), and their current–voltage characteristics were investigated for different temperatures. The samples showed good electrical parameters where n decreased from 2 for 80 K to 1.01 for 420 K. The barrier height $q\phi_{b}$ grew abnormally from 0.46 eV at 80 K to 0.83 eV at 420 K, due to the tunnel mechanism TFE effect. In addition, the samples showed low R_s_ which dropped from 13.80 Ω at 80 K to 4.26 Ω at 420 K. Finally, the results strongly suggested that the GaN thin layer caused an inhomogeneous barrier height, which was also in agreement with our previous simulations [5].

## Figures and Tables

**Figure 1 materials-14-05909-f001:**
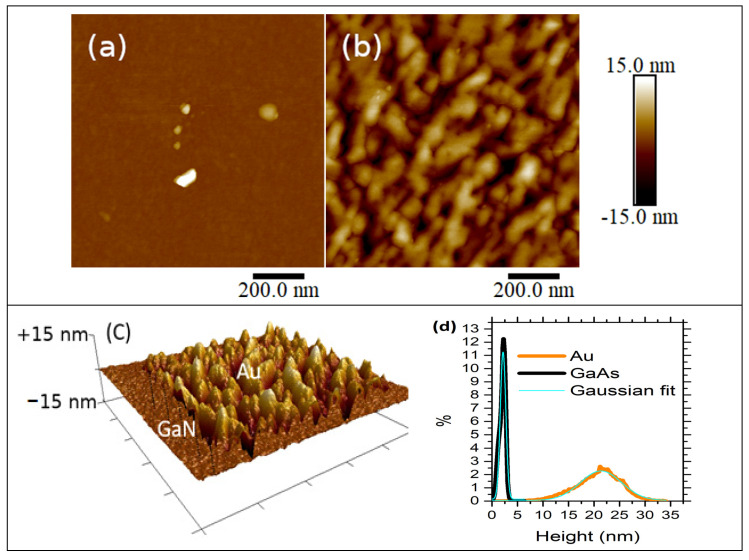
PeakForce tapping AFM topography for (**a**) GaN, and (**b**) Au surfaces. (**c**) Rendered illuminated 3D AFM image of GaN/Au frontier. (**d**) Height distribution functions obtained from the topography images.

**Figure 2 materials-14-05909-f002:**
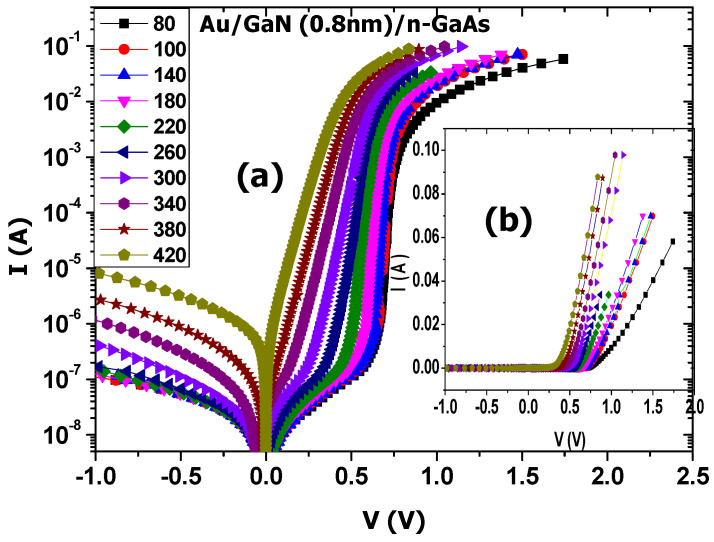
I−V measurements of Au/0.8 nm–GaN/n–GaAs structure, (**a**) semi-logarithmic scale and (**b**) linear scale.

**Figure 3 materials-14-05909-f003:**
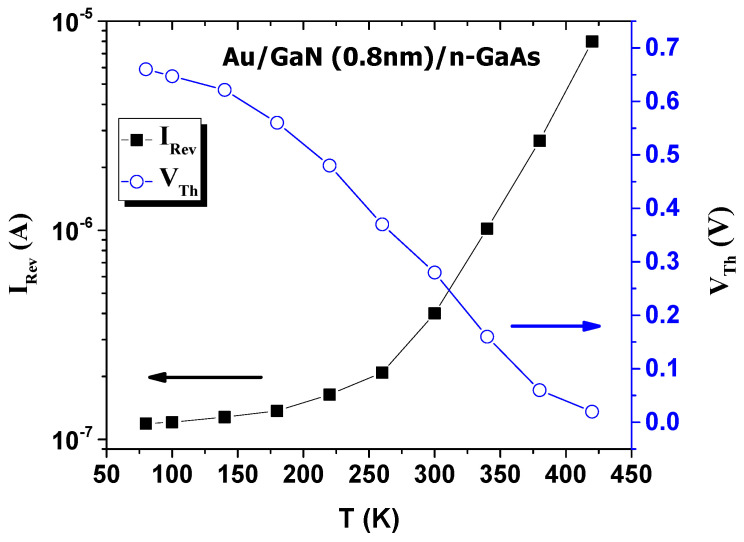
Variation of reverse leakage current and threshold voltage versus temperature.

**Figure 4 materials-14-05909-f004:**
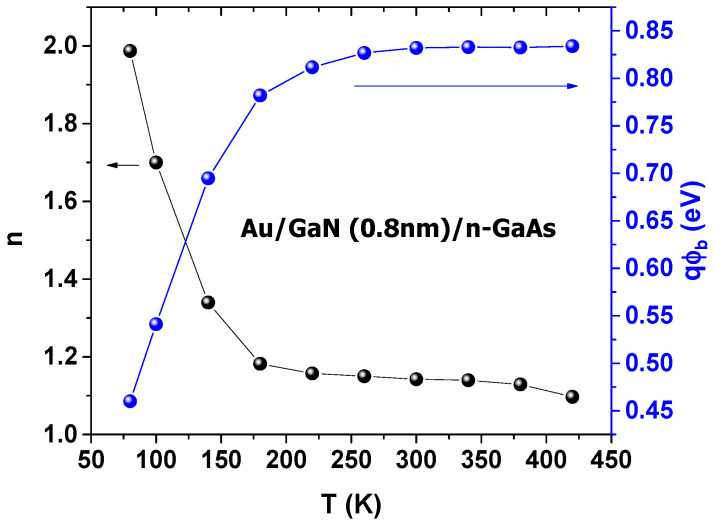
The *n* and $\phi_{b}$ extracted at each temperature.

**Figure 5 materials-14-05909-f005:**
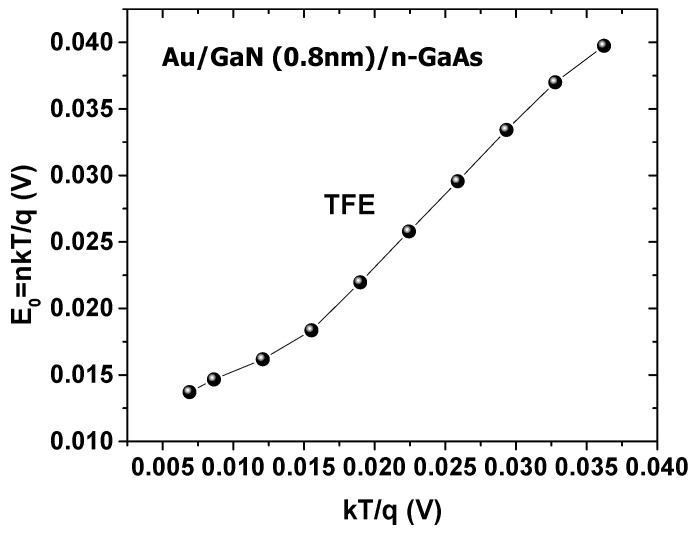
Variation of *E*_0_ (*nkT/q*) versus *kT/q*.

**Figure 6 materials-14-05909-f006:**
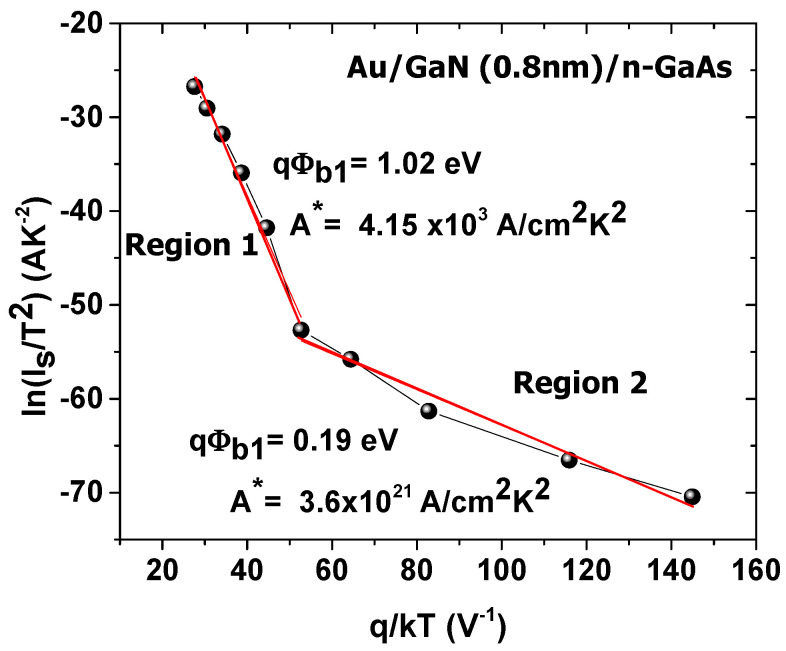
Variation of Richardson characteristic *ln*($I_{s}/T^{2}$) versus *q/kT*. Red lines are linear fits of experimental data according to the root mean square method.

**Figure 7 materials-14-05909-f007:**
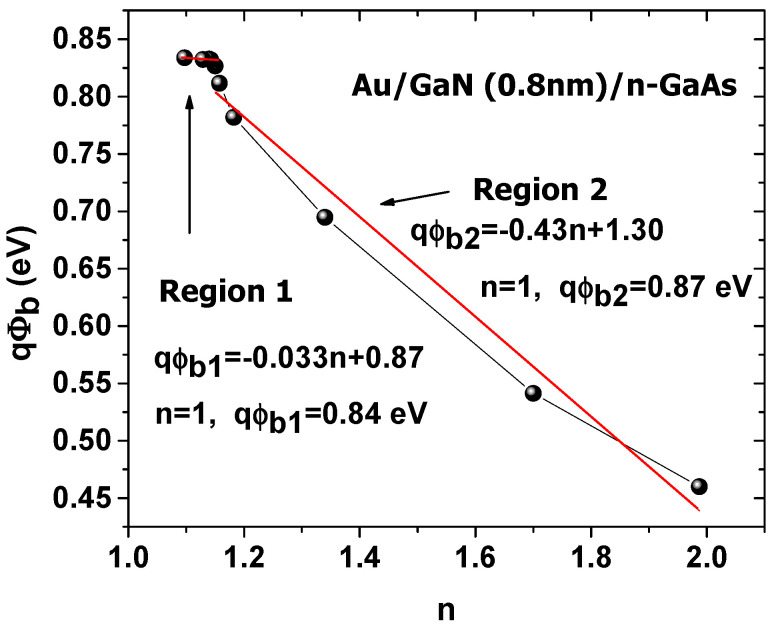
*ϕ**_b_* versus *n*.

**Figure 8 materials-14-05909-f008:**
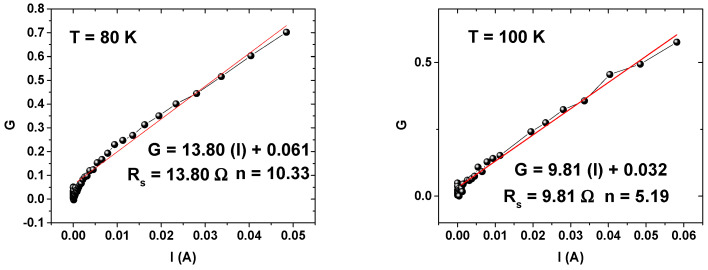
*dV/d(lnI)* plots at different temperatures.

**Figure 9 materials-14-05909-f009:**
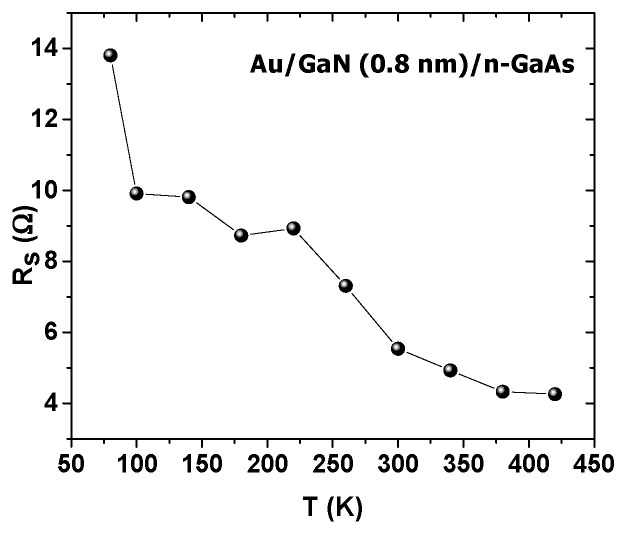
*R_s_* (Cheung and Cheung method) versus temperature.

**Figure 10 materials-14-05909-f010:**
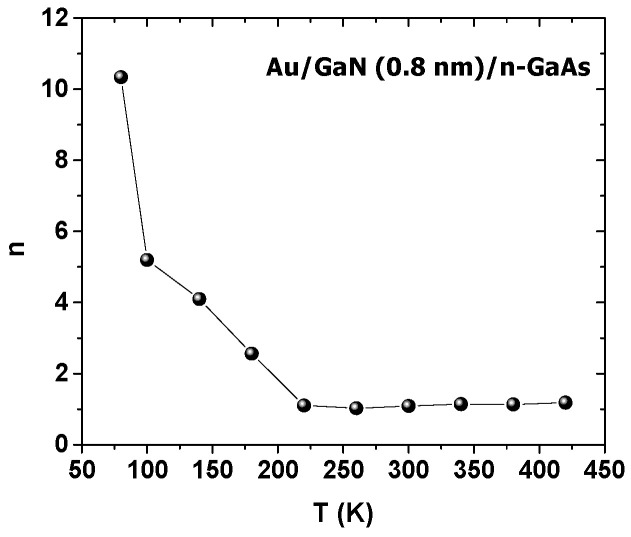
*n* (Cheung and Cheung method) versus temperature.

## Data Availability

The data presented in this study are available on request from the corresponding author.

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
