# Peer review of "Conduction Mechanisms in Au/0.8 nm–GaN/n–GaAs Schottky Contacts in a Wide Temperature Range"

_materials, 2021, doi:10.3390/ma14205909_

Round 1
Reviewer 1 Report
In this work, Au/0.8nm-GaN/n-GaAs Schottky diodes are studied electrically in a wide temperature range. With increasing temperature, the reverse current and barrier height increase. While, the ideality factor and series resistance decrease. The abnormal behaviors and their causes have been studied. Some recommendations are as follows.
(1)The roughness scale need tobe marked in Figure 1 (c).
(2)The value conditions of reverse leakage and threshold voltage shall be given.
(3)The units of resistance in Figure 8 need tobe marked.
Author Response
Thank you for accepting to review our paper and for your efforts and important comments which improved the quality of the manuscript. Corrections in the manuscript are highlighted in yellow.
Note: we have added a new figure (figure 3), and the figures numbers from 3 to 9 in the first version of the manuscript have changed by figure number +1 in the new version.
Comments and Suggestions for Authors,
In this work, Au/0.8nm-GaN/n-GaAs Schottky diodes are studied electrically in a wide temperature range. With increasing temperature, the reverse current and barrier height increase. While, the ideality factor and series resistance decrease. The abnormal behaviors and their causes have been studied. Some recommendations are as follows.
(1)The roughness scale need to be marked in Figure 1 (c).
Answer: the height scale has been added to Fig. 1(c).
(2)The value conditions of reverse leakage and threshold voltage shall be given.
Answer: we had added a new figure (figure 3) showing the variation of the reverse current at -1 V and of the threshold voltage with temperature.
(3)The units of resistance in Figure 8 need to be marked.
Answer: The unit of resistance in Figure 8 has been marked.
Reviewer 2 Report
The manuscript investigates Au/0.8nm-GaN/n-GaAs Schottky diode in a wide temperature range experimentally. Several parameters of the diode are extracted, and mechanisms responsible for their abnormal behavior are proposed. The experimental procedure is described correctly, while the introduction part and extracted results need additional clarification. Also, the main contribution of the manuscript has not been outlined.
- Introduction: Please specify which electrical properties of the Schottky diode should be improved by the nitridation of the interface.
- Page 3: Please comment on the correlation between the Schottky barrier height inhomogeneity and Au and GaN surfaces roughness, if there is one.
- Eqs. (1)-(5): Please list all the assumptions adopted for obtaining (3) from (1). It seems that you have neglected Rs. What value of the effective Richardson constant was used? Why have you applied the procedure from [45] only for obtaining Rs, but not simultaneously for n and qfb? Also, pay attention to the usage of qfb or fb in equations and eV and V as units in the whole manuscript (energy or electric potential).
- Page 6, below Fig. 3: Value of n at 420 K from Fig. 3 is 1.1 (not 1.01).
- Fig. 4: Please specify the units. Also, comment on the conclusions from this figure more extensively, not just by giving the reference.
- Fig. 5: Please revise the units of variables and usage of lowercase and uppercase k. What is the correct value of the Richardson constant in region 2? The values in the figure and text are with different exponents.
- Fig. 6: Relation for fb1 should have a negative slope.
- Page 10, before Conclusion: The series resistance is the resistance of the diode substrate and, in my opinion, not influenced by the quality of the interface (maybe by the annealing temperature). Please comment on the discrepancies of the n values obtained by two methods. Why values of qfb were not obtained by the Cheung and Cheung method? (Related to question 3).
- What is the main conclusion? Is the nitridation desirable or not, since it has twofold, opposite effects?
Reviewer 3 Report
Paper of Hicham Helal et al., describes temperature dependent current-voltage characterization of Au/GaN/GaAs Schottky diodes. Based on J-V-T measurements authors determined several metal-semiconductor contact properties and finally assigned thermionic field emission as a main conduction mechanism. Paper is very interesting for III-V semiconductor device community. However, the present version of the paper needs some improvement before the publication:
- Experimental part - I would recommend to use electron concentration n=4.9×1015 cm-3 instead of doping density. Density is rather a term which describe some properties related with area while concentration is related with the volume (see units of carrier concentration)
- Experimental part - Did authors confirm the GaN layer thickness and crystal quality using any other method, for example TEM or XRD measurement?
- Results part - the obtained RMS value for GaN layer of 0.6 nm is very small and compared with GaN layer thickness of 0.8 nm. In my opinion this needs some more discussion in the main text
- Figure 1 d - is the Y scale correct? I mean the order 0 to 0.8% or maybe 0 to 80%?
- Results part - how authors define reverse current and what is a difference between reverse current and saturation current?
- Figure 3 - please rotate right (blue) Y axis description by 180 degree
- Figure 9 vs Figure 3: why there is so big difference in the case of ideality factor n determined by Cheung method (Figure 9) and using equation 3 (Figure3)?
- In general, please take a look for some recent papers where authors investigate GaAs based devices by means of J-V-T measurements and conclude about conduction mechanism (TFE mainly):
Özavcı, E.; Demirezen, S.; Aydemir, U.; Altındal, S. A detailed study on current–voltage characteristics of Au/n-GaAs in wide temperature range. Sens. Actuators Phys. 2013, 194, 259–268, doi:10.1016/j.sna.2013.02.018.
Dawidowski, W.; Ściana, B.; Bielak, K.; Mikolášek, M.; Drobný, J.; Serafińczuk, J.; Lombardero, I.; Radziewicz, D.; Kijaszek, W.; Kósa, A.; Florovič, M.; Kováč, J., Jr.; Algora, C.; Stuchlíková, L. Analysis of Current Transport Mechanism in AP-MOVPE Grown GaAsN p-i-n Solar Cell. Energies 2021, 14, 4651. https://doi.org/10.3390/en14154651
Uslu, H.; Bengi, A.; Çetin, S.; Aydemir, U.; Altındal, S.; Aghaliyeva, S.; Özçelik, S. Temperature and voltage dependent current-transport mechanisms in GaAs/AlGaAs single-quantum-well lasers. J. Alloy. Compd. 2010, 507, 190–195. doi:10.1016/j.jallcom.2010.07.152.
Round 2
Reviewer 2 Report
The authors have improved their manuscript according to the reviewers' comments. Only a few remarks:
- The series resistance Rs doesn't change with the bias voltage significantly. At the low bias voltage V, there is a low value of the current I, and therefore the term I·Rs is low comparing to V which justifies the use of the relation (3) instead of (1).
- Fig. 6 needs units on the x-axis.
Reviewer 3 Report
I appreciate authors corrections. In my opinion current version of the paper is publishable in Materials.
Author Response
Dear reviewer,
Thank you again for accepting to review our paper and for your time and great efforts,
my best regards,